# High-volume, label-free imaging for quantifying single-cell dynamics in induced pluripotent stem cell colonies

Anthony J. Asmar[1]*, Zackery A. Benson[1], Adele P. Peskin[2], Joe Chalfoun[2], Mylene Simon[2], Michael Halter[1], Anne L. Plant[1]

**1** Biosystems and Biomaterials Division Material Measurement Lab, NIST Gaithersburg, Gaithersburg, Maryland, United States of America, **2** Software and Systems Division Information Technology Lab, NIST Gaithersburg, Gaithersburg, Maryland, United States of America

☯ These authors contributed equally to this work.
* anthony.asmar@nist.gov

**Data Availability Statement:** All relevant data are available at: https://doi.org/10.18434/mds2-2960 https://github.com/usnistgov/WSDOM.

## Abstract

To facilitate the characterization of unlabeled induced pluripotent stem cells (iPSCs) during culture and expansion, we developed an AI pipeline for nuclear segmentation and mitosis detection from phase contrast images of individual cells within iPSC colonies. The analysis uses a 2D convolutional neural network (U-Net) plus a 3D U-Net applied on time lapse images to detect and segment nuclei, mitotic events, and daughter nuclei to enable tracking of large numbers of individual cells over long times in culture. The analysis uses fluorescence data to train models for segmenting nuclei in phase contrast images. The use of classical image processing routines to segment fluorescent nuclei precludes the need for manual annotation. We optimize and evaluate the accuracy of automated annotation to assure the reliability of the training. The model is generalizable in that it performs well on different datasets with an average F1 score of 0.94, on cells at different densities, and on cells from different pluripotent cell lines. The method allows us to assess, in a non-invasive manner, rates of mitosis and cell division which serve as indicators of cell state and cell health. We assess these parameters in up to hundreds of thousands of cells in culture for more than 36 hours, at different locations in the colonies, and as a function of excitation light exposure.

## Introduction

Induced pluripotent stem cells (iPSC) can be created from any adult cell, they can expand almost indefinitely, and they can be induced to differentiate into any cell type of the body. iPSC therapies are being developed to treat diseases, reconstruct tissues, and screen drugs [1]. The culture, expansion, and differentiation of iPSCs in either a research or manufacturing setting is challenging because of their sensitivity to conditions such as media composition, physical handling (such as passaging protocols), and the variations that exist between different cell lines. This presents the need for better quantification, and real-time monitoring, of the effects

**Funding:** The author(s) received no specific funding for this work.

**Competing interests:** The authors have declared that no competing interests exist.

of culture conditions on stem cell renewal, expansion, and maintenance of pluripotency. Time-lapse imaging of live cells is a powerful quantitative tool for longitudinal measurements of iPSCs.

Fluorescence labeling is a powerful tool in cell imaging and in the creation of image analysis algorithms, but exposure of cells for long times to potentially harmful excitation irradiation is undesirable [2–6]. The non-destructive nature of low-intensity transmitted light imaging allows for real-time, relatively non-invasive, optical probing of living cells in culture over long times, but thus far it has not been possible to segment and track individual IPSC by transmitted light imaging alone. Furthermore, if adequate analytical pipelines can be established and validated using low-intensity transmitted light, any cell line can be quantitatively imaged over time in an non-invasive manner. Metrics such as morphology, migration rate, division times, and expression of reporter molecules could be used for characterizing and monitoring the state of cells in real-time during culture and expansion processes. Optical imaging that provides such metrics would allow the testing and development of unique culture and expansion conditions that are optimal for different cell lines. The spatial and temporal nature of live cell imaging provides contextual knowledge that is not available from single timepoint and non-imaging measurements.

The growth of pluripotent cells in multicellular colonies has posed challenges to quantitative imaging at the cellular level because of their close cell-cell contacts. Imaging at the colony level [7] or as groups of cells [8] in brightfield and fluorescence have led to knowledge about stem cell colony growth characteristics, gene expression, and classification at different stages of stem cell culture. Neural network-based algorithms have enabled greater discrimination of the characteristics of groups of iPSC and have been successfully used to predict differentiation [9] or distinguish between iPSC colonies, differentiated cells and dead cells from phase contrast images in an automated expansion culture setting [10–12]. In another study, an assessment of individual iPSC nuclei based on a fluorescent label and an ensemble of neural networks addressed the challenge of dense cell tracking and localization of each cell nucleus in an iPSC colony but required the signal of a fluorescent probe [13]. Recently, Atwell et al. reported the development of a 3D U-Net to segment individual cells in 3D iPSC cultures imaged in bright field; this model was also trained with large numbers of nuclei (250000) through a combination of manual and model-dependent segmentations [14]. We present here quantitative dynamic imaging and tracking of unlabeled individual iPSCs and their progeny over time.

We previously developed a neural network-based workflow for segmentation of single iPSC nuclei from label-free, transmitted light microscopy images [15]. The training data did not require manual annotation but rather utilized a reporter cell line expressing a green fluorescent protein (GFP)-nuclear protein fusion product (LaminB-GFP) to enable nuclear segmentation. This approach for automated segmentation allowed generation of large training data sets and an evaluation of model performance as a function of the amount of training data. Inferenced nuclear detection incurred false positive rates and false negative rates of 0.16 and 0.19, respectively. More recently we developed a 3D neural network mitosis detection model for dividing iPS cells from label-free, transmitted light microscopy images [16]. The model generation depended on manual annotations and a unique semi-supervised approach to iteratively expand the training data size and improve model performance. The final model successfully identified 37 of the 38 cell division events in manually annotated test image stack.

In this study, we build on our prior work to report a label-free, transmitted light acquisition and analysis workflow for segmenting and tracking individual iPSC nuclei, including mitosis detection for characterizing the onset of mitosis, division, and parent-daughter cell linkages. This workflow required refinement of our previous models to improve performance, and the

use of a cell line expressing a different fluorescent nuclear probe, a fusion protein of green fluorescent protein (mEGFP) and the histone protein, HIST1H2BJ, which enabled improved mitosis detection for annotating phase contrast images for training the models. Both training and evaluation of the models rely on large datasets of auto-segmented images of fluorescent nuclei.

Ultimately, we envision using this analysis pipeline to enable the label-free tracking and quantification of reporters of gene expression in individual cells over time. Because the complexity and stochastic nature of cell regulatory pathways results in heterogeneous responses from different cells, it is critical to collect data on statistically sufficient numbers of individual cells, and this requirement precludes heavily supervised image analysis. Here our goal is to minimize the use of human intervention while efficiently collecting large sets of data for training and evaluating the segmentation and tracking of label-free iPS cells.

## Methods

### Cell culture

A genetically engineered cell line expressing green fluorescent protein (mEGFP) covalently fused to the histone protein, HIST1H2BJ, was developed by Allen Institute for Cell Science (mEGFP-tagged Histone H2B type 1-j) and procured from Coriell (AICS-0061-036, Philadelphia). Parental iPSCs (WTC-11, GM25256, Coriell) and hESCs (WA09, WiCell, Madison WI) were maintained under feeder-free conditions on Matrigel-coated plates (#354277, BD Biosciences) in mTeSR™ Plus (#100–0276, StemCell Technologies Inc.). iPSCs were routinely passaged with Accutase (# 07920, StemCell Technologies Inc.), while hESCs were routinely passaged with collagenase (#07909, StemCell Technologies Inc.). For experimental setups, cells were passaged as single cells using Accutase, enumerated using a Multisizer 3 Coulter Counter (Beckman,), and then seeded at the desired density in 6-well tissue culture plates (TPP, Trasadingen, Switzerland). U-Net models were trained using data with cells seeded at 5,200 cells/cm$^2$ (low-cell-density), 10,400 cells/cm$^2$ (medium-cell-density), and 20,800 cells/cm$^2$ (high-cell-density). All live cell experiments were performed on cells seeded at approximately 10,400 cells/cm$^2$. Spy595-DNA (CY-SC301; Cytoskeleton, Denver CO) was used to label nuclei for generating reference data for WTC-11 and H9 cells according to manufacturer's instructions.

### Image acquisition workflow

This work was performed in our Wide Scale Digital Optical Microscopy (WSDOM) laboratory. Zernike phase and fluorescence images were collected using a 10X/0.3 numerical aperture objective (Zeiss part number 420341-9911-000) on a Zeiss Axio Observer.Z1 microscope (Carl Zeiss USA, Thornwood, NY) equipped with motorized x-y stage (SCANplus IM 130x80, Marzhauser, Germany) and an ORCA-Fusion BT Digital CMOS camera (C15440-20UP, Hamamatsu, Japan) for image capture. Images were typically collected at 2min intervals. For each time point, multiple images are acquired (2000x2000 pixels per field of view) with 10% overlap between tiles. Each field-of view contains 7 phase contrast images at different z-planes and a single fluorescence image. A spatial calibration target was used to determine that each pixel is equivalent to an area of 0.406 μm$^2$. During automated acquisition, the microscope and all peripherals were controlled with an electronics and software controller system that enabled faster data acquisition than is typically achieved with instrumentation software provided by most manufacturers (Inscoper, Rennes, France). A detailed description of the acquisition protocols and benchmarking of the microscope system is provided in the S1 Text. A description of experimental datasets including the experiment name, light exposure, cell counts, number of mitoses, and length of experiment, see S1 Table.

## Generating nuclear object reference data

Typical datasets include hundreds of images in a time-lapse acquisition, where each stitched, single timepoint image contains tens of thousands of cells. This volume of data presents a sampling challenge for assessing the performance of a label-free nuclear segmentation algorithm. Manually annotating thousands of cells would yield less than a 0.1% sampling rate. To address this big data challenge, we used classical segmentation routines to automate the creation of reference objects from GFP fluorescence images for testing the performance of our U-Net models.

The reference masks for training the 2D U-Net are generated by segmenting the GFP fluorescence image using a band-pass filter followed by an empirically determined intensity threshold value. To separate touching objects, the mask objects are segmented with a watershed-like segmentation routine, FogBank [17] followed by a binary erosion.

## Training the 2D U-Net

We trained a 2D U-Net to segment single-cell nuclei from phase contrast images starting with a pre-trained U-Net [15] as our initial network. Training data consisted of a 15000x15000 pixel subset of a large-scale single timepoint stitched phase contrast image (approximately 230 million pixels), and a corresponding fluorescence image where nuclei are labeled with GFP.

The phase contrast images are normalized using z-score normalization, and the final images and reference masks are tiled into 256x256 tiles to match the input of the U-Net. The relative weights of the foreground and background classes is 2 to 1.

Image augmentation was used during the training process in which the input training data were modified by operations such as reflection, rotations, and by applying Gaussian blur to mimic potential variations in cell orientation and focus. Training of the U-Net was done in MATLAB (https://github.com/usnistgov/WSDOM).

## Inferencing for label-free segmentation

The trained U-Nets operate on normalized phase contrast images and return binary foreground/background masks corresponding to single-cell nuclei. The binary masks are created by inferencing with 3 instances of the same model and thresholded by 2 (as explained in more detail in S2 and S3 Figs). The binary masks are post-processed using FogBank segmentation [17] to separate touching objects. Fogbank includes two input parameters, the erosion size and the minimum size. The effect of these parameters on nuclear detection performance was evaluated (S1 Fig). We chose parameters that maximized the performance for the rest of this study.

## Assessing the accuracy of label-free segmentation

The accuracy of inferenced segmentation was calculated after using a linear sum assignment routine with the cost function proportional to both the centroid distances and the overlaps between the segmented GFP reference data and the inferred test set. The distance cutoff between two objects is 15 pixels. The overlap is computed as the intersection over the union of the two objects. The number of false positives or over-segmented objects are normalized by the reference cell count and reported as fraction of additional objects, and the number of false negatives or under-segmented objects are normalized and reported as the fraction of missing objects. We also report F1 scores which were calculated as

$$F1 = \frac{2\text{TP}}{2\text{TP} + \text{FN} + \text{FP}}$$

as in [18], where TP is number of true positives, FN is number of false negatives, FP is number of false positives.

## Evaluating the accuracy of automated segmentation for generation of reference data

To estimate the accuracy of automated segmentation of fluorescent nuclei that were used for training and evaluation, we manually annotated centers of nuclei in a small number of fluorescent images. Comparison of automated and manual reference data are discussed in Results and in S4 and S5 Figs. Errors in object identification were calculated after using a linear sum assignment routine with a cost function proportional to the center distances between the manual annotations and reference fluorescence masks. By comparing manually detected nuclei to nuclei detected with the automated segmentation routine we determined that, while most manually annotated fluorescent nuclei are concordant with automated segmentation results, there were occasionally nuclei that were difficult to detect, where both manual and automated annotation resulted in ambiguous results (S4 Fig). By comparing automated reference data with manual data, we determined that the reference data are highly accurate, although some apparent errors occur, mainly due to the difficulty in segmenting overlapping nuclei, resulting in missing objects (S5 Fig). The presence of errors in the reference data means that the error rates associated with the inferenced results is slightly overestimated. Nonetheless, the similarity between the reference data and the manually annotated data provides confidence that the reference data produced by automated segmentation can be used to compare and evaluate AI models for label-free segmentation while providing a nearly limitless number of objects that can be sampled.

## Creating training data for label-free mitosis detection

To prepare the training data for the U-Net, GFP data are used to segment cell nuclei for the 2D U-Net with class 0 to designate background and class 1 for nuclei (see *Training the 2D U-Net* section above). Potential mitotic nuclei were then labeled by applying an additional band-pass filter to identify the condensed DNA in the original GFP image followed by a threshold. The pixel values of these objects were designated as class 2. Finally, a tracking routine was used to identify daughter cells. Each object designated as class 2 in frame t_0 is duplicated and then a linear sum assignment routine pairs that object with an object in the subsequent frame t_1. If a cell in t_0 maps to two objects in t_1, then the initial cell is designated as class 2, and the two mapped cells are designated as class 3. A final postprocessing step is applied to designate the object as class 2 (i.e., a mitotic nuclei) in each of the 5 frames before division, and as class 3 (one or two daughter cells) in each of the 3 frames after division. The resulting three-dimensional data were tiled into 128x128x16 images for input into the 3D U-Net.

## Training the 3D U-Net for label-free mitosis detection

To identify mitosis, we built on a previously published 3D U-Net [16] and modified the training data to output a 4-class mask (0 –background, 1 –nuclei, 2 –mitotic nuclei, 3 –daughter cells). For the 3D U-Net, the first two dimensions are the x, y pixels of the image, and the third dimension is consecutive time frames (two-minute separation between frames). The phase contrast images are normalized using z-score normalization, and the final images and reference masks are tiled into 256x256x16 tiles (in the x, y, z directions). During augmentation, a 128x128x16 section of that tile is randomly selected across the x and y directions to match the input of the U-Net. Additional image augmentation was used during the training process in which the input training data were modified by reflections, rotations, and by applying

Gaussian blur. The relative weights of the background, non-mitotic nuclei, mitotic nuclei, and daughter cells are 1 to 2 to 20 to 20. Training the 3D U-Net is done in Python using Tensor-Flow and Keras (https://github.com/usnistgov/WSDOM).

## Post processing of 3D U-Net inferenced results

Mitotic events are detected by combining class 2 and 3 objects as one class–i.e., removing objects that are not mother or daughter nuclei–and counting the number of class 2 and class 3 objects in three dimensions (x, y, time). Potential false positives are filtered by requiring each object to have a volume larger than 300 pixels and persist for at least 10 frames. For subsequent tracking of daughter cells for each event, class 3 designation was used, and the time of mitosis was determined as the time when two objects become identified or when the class 2 object is the largest. If only one class 3 object is identified, then two daughter nuclei are labeled at the major axis positions opposite the mother cell label.

## Evaluation of the accuracy of the 3D U-Net for mitosis detection

Mitoses detected from the 3D U-Net were compared to reference data of 4-class segmentation of GFP fluorescence (see *Training the 3D U-Net* subsection above). To estimate the accuracy of the 4-class automated segmentation results, we manually annotated 629 mitoses across two experiments. The manual data was paired to the 3D U-Net inferenced results using a linear sum assignment routine with the cost function being proportional to the distance between mitosis events in space with an empirically determined spatial cutoff of 15 pixels and a time cutoff of 6 frames.

## Cell tracking with mitosis detection

The inputs for cell tracking are cell nuclei detected using the 2D U-Net, and parent/ daughter positions obtained from the output of the 3D U-Net. Cells are tracked using a tracking routine written in Python [19, 20]. The tracking uses a cost function that weights the overlaps as well as centroid distance between objects in consecutive frames to link or map cells across time. In order to reduce the loss of true objects, any unmapped objects are checked for merge events, and any tracked object can remain unmapped for 5 frames after which the track is considered lost. This can result in the appearance of additional segmented objects for single timepoints. Dividing parent/daughter positions are used to track cell lineage and are omitted from the linear sum assignment routine which is applied only to track non-mitotic nuclei.

## Evaluation of cell-tracking workflow

We evaluated our label-free tracking workflow using GFP reference data. To calculate the accuracy, we paired the tracked objects between our reference data and test set (x, y, t, id) in each frame using a centroid-distance cost function as used for the evaluation of the segmentation and mitosis detection. We then calculated the linkage-error rate for each track. A linkage error is when a label-free track id changed, and the reference id did not. The linkage error rate per track is then the sum of these errors divided by the length of the track.

Another intuitive technique to assess tracking accuracy is to compare the long-time measurements between different segmentation models. The metrics that we used are the track lengths (the total number of frames a cell is continuously identified) and the interdivision times (the time between appearance of a daughter cell and its subsequent division), in which a more accurate tracking sequence should have longer track lengths as well as interdivision

times that are biologically relevant (i.e., between 9 and 24 hours). We checked the accuracy of the GFP reference data with a small amount of manual annotations.

## Data availability

The data that support the findings of this study are openly available in the NIST Data Repository at https://doi.org/10.18434/mds2-2960. All code and models generated are available at https://github.com/usnistgov/WSDOM.

## Results

The goal of this study is to develop a rapid and robust imaging pipeline that enables in-depth quantitative analysis of unlabeled nuclei in live iPSCs imaged by phase contrast microscopy. Here we develop and evaluate U-Net models that allow segmentation of individual nuclei within iPSC colonies; we also detect and track mitotic cells and the formation of their daughter cells. The output masks of the U-Net models are processed to provide input for tracking and lineage construction of cells for the length of the acquisition.

The general progression of the workflow, as described by the schematic in Fig 1, takes in unlabeled time-lapse transmitted light microscopy images. Two different U-Net operations are applied to the data. The first is inferencing with a 2D U-Net followed by object separation to detect the individual nuclei in each image. The second is inferencing with a 3D U-Net which takes in time-lapse images acquired at 2 min intervals to detect and label mitotic events as well as parent and daughter nuclei. The segmented objects are automatically tracked to link the objects over time and are combined with the labels from the 3D U-Net to generate cell lineages.

Because of the spatial proximity of cells in pluripotent cell colonies, segmenting and tracking individual cells is challenging. Mitosis event detection is also challenging. Adherent mammalian cells in culture frequently round up during division resulting in a high contrast image patterns that can be observed and automatically detected [21]. The iPSCs have a more subtle appearance during division, remaining flat with a uniform contrast pattern throughout division. In this work, we have used a mEGFP-HIST1H2BJ cell line to provide fluorescent nuclei that can be segmented and used for training the 2D and 3D U-Nets. High signal-to-noise ratio

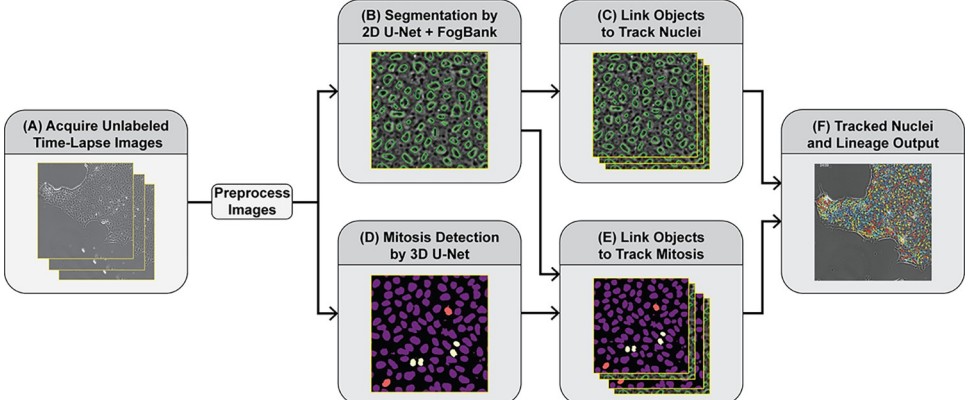

**Fig 1. Schematic of the workflow progression for the analysis of time lapse Zernike phase images.** (A) Tiled, time-lapse images are acquired and pre-processed by focal plane selection and image stitching. (B) Stitched phase contrast images are inferenced with a 2D U-net, then object separation is performed with the Fogbank algorithm for nuclear segmentation. (C) Nuclear objects are tracked. (D) Time-lapse, stitch phase contrast images are inferenced with a 3D U-Net for mitosis event detection and daughter nuclei identification and detection. (E) Tracked objects from (C) are linked with mitosis events and daughter nuclei from (D). (F) Tracked nuclei and lineages are output. See Methods and S1 Text for details.

images can be acquired using this probe allowing for high quality nuclear segmentation with classical image analysis routines. This approach provides the advantage of reducing or eliminating the need for manual segmentation, which in turn allows us to sample a very large volume of reference segmentations for evaluation of the label-free analysis workflow. This latter point is critical because the heterogeneity of cell characteristics and responses demands a large statistical sampling, and this has not typically been accessible particularly in the temporal domain.

The process for development of the 2D U-Net is shown in the schematic in Fig 2A and the steps are described in Methods. Inferenced results were compared to segmented fluorescent images to evaluate the accuracy of the model.

## Evaluation of 2D U-Nets

**Comparing models trained with different densities of cells.** We created models from three sets of training data containing cells plated at different densities (example images shown in Fig 2B). Depending on the density of cell seeding, the time that cells are in culture and the natural variability of cultured cells, iPS cells can assume subtle differences in morphologies due to differences in interaction with neighboring cells and adhesion to substrate. To increase the likelihood of producing a model that is applicable for different datasets and over the times that cells are in culture, we trained three models with images from three different wells of cells that were prepared by seeding each well with a different number of cells. We refer to these 3 models as low-cell-density, medium-cell-density, and high-cell-density models. The models were tested against images at different densities of cells to evaluate the generalizability of the models for different cellular contexts. In Fig 2C we show a comparison of the performance of those 3 models with image data that were collected on cells over 20 h in culture. The F1 scores, fraction of additional objects and fraction of missing objects were calculated for single frame images taken at different times over 20 h of imaging. The data are plotted as a function of the density of cells (number of cells per $mm^2$) in the frame. As shown, the models perform similarly over the entire 20 h experiment, whether cells are at relatively low densities or if cells are at relatively high densities. Comparison of the fraction of additional objects, the fraction of missing objects and the F1 score between 2D U-Net models showed that the performance of the 3 segmentation models was not highly sensitive to cell density, and the high-cell-density model performed nominally better than the others with the 'fraction of additional objects' being the largest performance difference between the models. The high-cell-density model produced an error fraction in this dataset of approximately 0.05 to 0.08 additional objects, and a fraction of approximately 0.03 to 0.06 missing objects, with an F1 score that ranged for 0.92 to 0.95 over the time course of the measurement. While the difference in performance of the 3 segmentation models is negligible, a distinction becomes apparent when cells are tracked over long times, as will be discussed later.

**Comparison of different datasets.** We examined different datasets to assess the performance of the high-cell-density model on nominally identical cell preparations. The data in Fig 2D show the accuracy metrics for 5 cell samples from 3 different days, examined over 13 h in culture. The data are plotted to indicate the model performance in each frame collected over time and color-coded to indicate the density of cells in the frames. The average F1 score over all 5 datasets is 0.94. These data suggest good concordance in segmentation performance across within-day replicates and experiments on different days. It should be appreciated that many variables are at play when evaluating different datasets in addition to the performance of the U-Net, including unknown biological and experimental variables. The replicability of these data provides confidence that the analysis workflow, including the U-Net, can be robustly

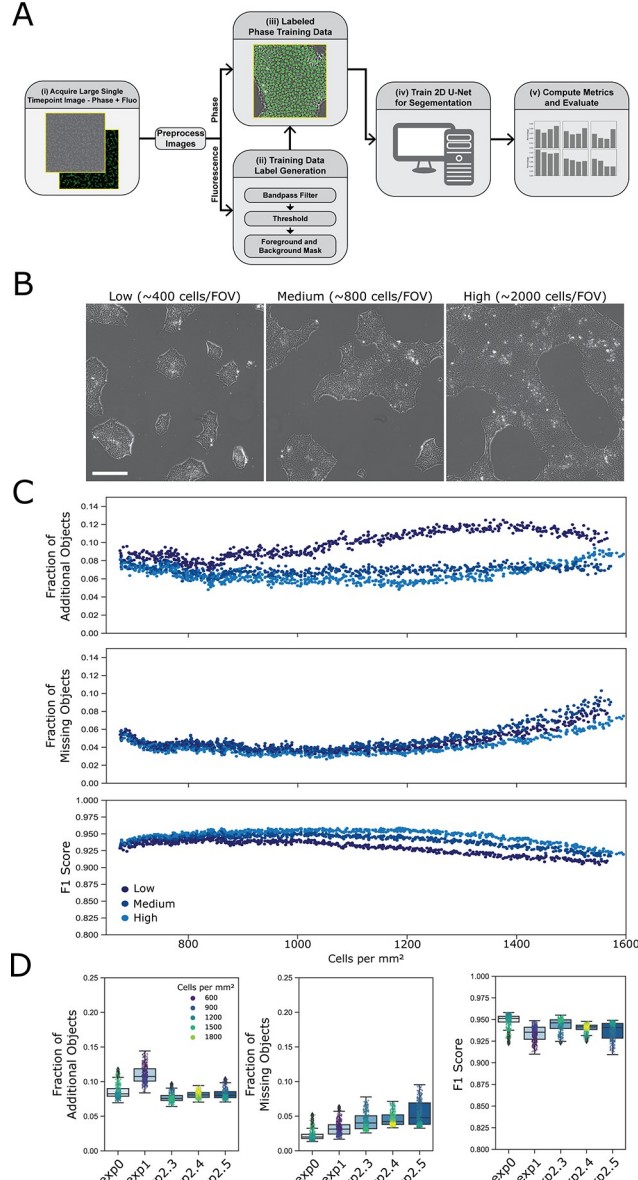

**Fig 2. Training and evaluation of 2D U-Nets.** (A) Schematic describing the process for model training and evaluation of 2D U-Nets for iPSC nuclear segmentation. (i) A single timepoint image set consists of a phase contrast image created by stitching 100 fields of view (FOV) and a corresponding fluorescence image where nuclei are labeled with GFP. FOV are acquired and pre-processed to identify the best in-focus z plane and then stitched (see Methods). (ii) The fluorescence images are processed to produce a foreground and background mask. (iii) The processed fluorescence image is combined with the unlabeled image for (iv) training the 2D U-Net. (v) The accuracy of inferencing individual nuclei from phase contrast images is evaluated by comparing the locations of inferenced nuclei with fluorescent nuclei in reference images. (B) Representative phase contrast images of images of cells seeded at relatively low, medium, and high densities. Scalebar shown is 250 µm. (C) The 'fraction of additional objects', 'fraction of missing objects' and the F1 score determined for each frame of one image dataset collected over 20 h. Accuracy metrics are plotted versus the measured number of cell nuclei/mm$^2$ per frame. The inferenced results were compared for the three models trained on either low (i), medium (ii), or high (iii) cell densities. (D) Using the high-cell-density model, the 'fraction of additional objects', 'fraction of missing objects and the F1 scores are plotted for five replicate datasets. The first dataset in the plot are the data shown in panel 2C. Each datapoint represents the corresponding error rate for that frame, and the dot color indicates the density of cells in the frame. Tukey box plots indicate summary statistics for each time-lapse dataset.

applied to nominally different datasets. Segmentation can be compromised by various aspects of the cell culture. These caveats are discussed in S1 Text.

To test the robustness of the segmentation model for different colony-forming cell lines, we applied the model to phase contrast images from the parental WTC-11 iPSC line and the embryonic stem cell line H9. These two cell lines are distinct from the cell line that was used for training the algorithm. The GFP-producing cell line used in training was derived from the WTC-11 line, but subtle differences in derived lines are often observed. We expected that the embryonic H9 cell line might be very different from the iPSC cells used for training. Again, subtle differences in cell and colony morphology seem apparent by eye. The scores for performance of the model when applied to these two cell lines were determined by comparing inferenced results with automated segmentation of reference data created by staining these cells with Spy595-DNA (see Methods). Model performance for these cell lines is similar to what was observed with the GFP-producing cell line, indicating model generalizability to these other pluripotent stem cell lines (data shown in S6 and S7 Figs). In addition, we compared F1 scores on data taken from 2 nominally different microscope systems and again found negligible differences (F1-scores of 0.95 and 0.94; see Supplemental Materials for details).

## Development and evaluation of mitosis detection with a 3D U-Net

A 3D U-Net model was developed to detect mitosis in phase contrast images using training data generated by automated segmentation of the histone-labeled GFP fluorescent nuclei (see schematic in Fig 3A). The result is a 3D U-Net with a four-label output as shown in Fig 3B and S1 Video consisting of background (class 0), non-dividing cell nuclei (class 1), nuclei undergoing mitosis (class 2), and daughter nuclei (class 3) immediately after division. The training data for the 3D U-Net were constrained by applying a class 2 label for 5 frames before a division event and applying a class 3 label for 3 frames after the corresponding division event as determined by analysis of the GFP fluorescence data. The output of the 3D U-Net was postprocessed to remove spurious mitotic events by filtering for size and duration. Details are provided in Methods.

The performance of the 3D U-Net model was directly evaluated by comparing inferenced results with automatically identified mitotic events from fluorescence reference images (Fig 3C). For a dataset consisting of 5004 reference mitotic events that were detected by the automated mitosis detection algorithm applied to the fluorescence data, the 3D U-Net resulted in 3164 true positive mitotic events, 796 additional events, and 1840 missing events for an F1 score of 0.70.

We also examined, with a much smaller amount of manual annotations, the performance of the automated reference data and the inferenced results from the 3D U-Net. Compared with 629 manually annotated mitosis events, the 3D U-Net inferencing resulted in an F1 score of 0.78. Comparing the results of manual annotation with the number of mitosis events detected from the fluorescence data, the F1 score was 0.87. These data indicate that while the fluorescence reference data were in good concordance with manual data, the 3D U-Net tended to miss mitotic events more frequently (182 out of 629 mitotic events were missed by the 3D U-Net). We conclude that the auto-generated reference data is reasonably accurate, and an F1 score for the 3D U-Net of 0.70 is a reasonable estimate. No spatial or temporal dependence of the performance of the mitosis detector was observed.

## Label-free tracking of single iPS cells

Time-dependent cell tracking can provide greater insight into evaluation of biological differences between samples and treatments. Additionally, time-dependent cell tracking allows us to

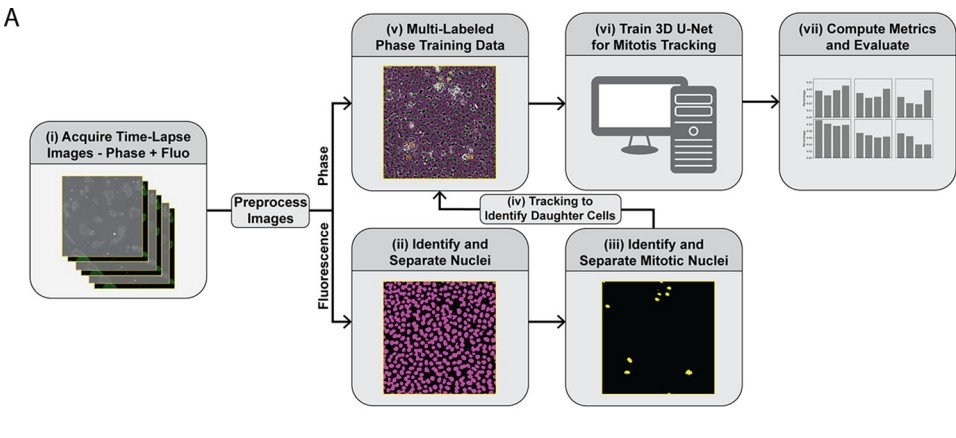

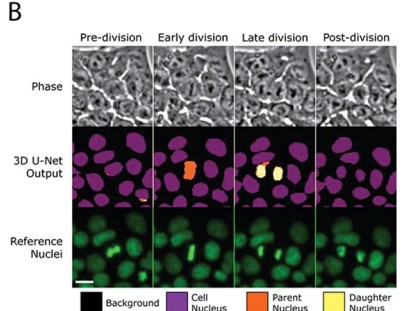

**Fig 3. Development and evaluation of mitosis detection with a 3D U-Net.** (A) Schematic describing the process for model training and evaluation of a 3D U-Net for iPSC mitosis detection. (i) A time-lapse image set consists of a phase contrast image created by stitching 4 fields of view and a corresponding fluorescence image where nuclei are labeled with GFP. Images are pre-processed to identify the best in-focus z plane and stitched (see Methods). (ii) The fluorescence images are processed to first identify and classify individual nuclei. (iii) the fluorescence images are processed again to identify and classify mitotic cells. (iv) Afterwards, the different classified cells are tracked over time to identify and classify the daughter cells. (v) The different classes of cells are then combined with the phase contrast images for (vi) training in the 3D U-Net. (vii) The performance is evaluated by comparing the locations of inferenced mitotic events from the phase contrast images with mitotic events identified in GFP fluorescence reference images. (B) Example of a mitotic event occurring in the phase contrast images along with the 3D U-Net classification output with its different class labels and the corresponding fluorescence reference images. Scalebar shown is 15 μm. (C) The calculated error in mitosis detection by the 3D U-Net inferencing when compared to manually annotated reference data, and the error in the fluorescence classification when compared to manually annotated reference data. Comparison of fluorescence reference data (GFP (Auto)) with inferenced (U-Net) data indicates that errors in inferencing are not due to errors in auto classification.

filter out spurious short-lived tracks to improve confidence in the data from accurately identified cells. From a biological perspective, the temporal phase contrast image data allow us to access single-cell dynamics in large cell populations over multiple cell doublings. The results of our label-free 2D U-Net segmentation and 3D U-Net mitosis detection routines were used in a cell tracking routine (as described in Methods) in order to follow individual cells and their progeny over time (S2–S4 Videos).

Filtering objects that track for only a short period of time (i.e. less than 5 frames or 10 minutes) and removing them from the dataset eliminates spurious objects and other false positive events (Fig 4A). The fraction of additional objects decreases as a function of minimum track length, which indicates that longer tracks are true positive segmentation events, i.e., the objects we are tracking are indeed cells. Filtering short tracks also has implications for cell counting. Removing tracks that are less than 10 minutes in duration results in a cell count that is more consistent with the count from our GFP reference dataset (Fig 4B). Thus, this filtering routine helps to establish greater confidence in both dynamic measurements and also static

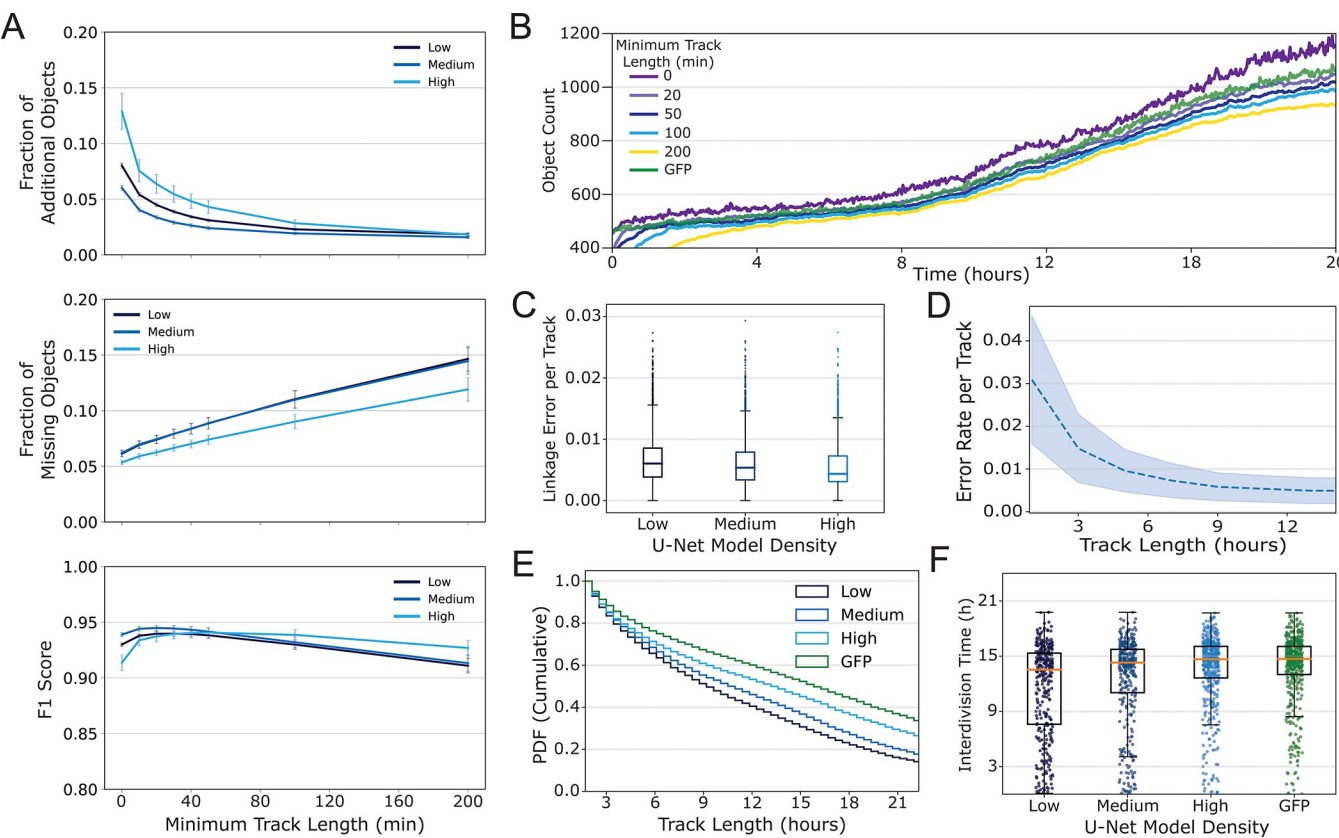

**Fig 4. Tracking of segmented iPS cells; comparison of the 3 different segmentation models.** Labels "High", "Medium" and "Low" refer to high-cell-density, medium-cell-density, and low-cell-density U-Net models. (A) After tracking, eliminating objects with short track lengths reduces segmentation errors. Including only objects that were successfully tracked for times greater than the minimum track length results in datasets with fewer segmentation errors. The high-cell-density segmentation model (which was created with training data produced with the highest number of cells) performed better overall with respect to fraction of missing objects and F1 scores than the other two models after track length filtering. (B) Examining inferenced results from the high-cell-density model indicates that eliminating objects that were tracked for less than 20 consecutive minutes appears to produce a more accurate cell count. The green line is the number of segmented fluorescent cells in a colony that were counted in the corresponding GFP image; that number is nearly identical to the inferred data when a filter of 20 sequential minutes is applied. (C) Tukey boxplot of Linkage Error Rate per track computed by comparing inferenced data from the high-cell-density, medium-cell-density and low-cell-density segmentation models with reference tracks created from GFP fluorescence data. Lines in boxes indicate the median error for inferenced tracks was slightly lower when the high-cell-density model was applied to the data. (D) Tracking error rate computed for the high-cell-density model as a function of track length. (E) Cumulative probability distribution of track lengths inferenced from the high-, medium-, and low-cell-density models and from auto segmentation of GFP. (F) interdivision times (hours from a division event to a subsequent division of those daughter cells) for cell tracks computed using the different cell-density models or auto GFP segmentation. Interdivision times that appear to occur within 9 hours are considered physiologically improbable and are likely to be tracking errors. Auto GFP segmentation resulted in 8 of these events compared to 91, 103, and 151 such events for the high-cell-density, medium-cell-density, and low-cell-density models. The numbers of interdivision times that were measured at >9 hours were 441 for the auto GFP segmentation, and 405, 291 and 228 for the high-cell-density, medium-cell-density and low-cell-density U-Net models.

measurements such as cell count. More stringent filtering criteria can lead to fewer objects tracked but provides greater confidence in the biological implications of the objects' behavior (S7 Fig).

We assessed track lengths as an estimate of the quality of our cell tracking workflow. We quantified cell-tracking accuracy with three different measurements: (1) the linkage error rate per track, (2) the track length distribution and (3) the interdivision times. Using the GFP fluorescence to generate reference tracks, the data in Fig 4C show the linkage error rate computed for an entire dataset (4000 initial cells) to be approximately 0.005 on average, which corresponds to one potential linkage error every 8 hours on average over the course of a single track. Approximately 70% of nuclei within a population can be tracked without linkage errors

for more than 8 h. Linkage error rates occur most often in regions of colonies that are difficult to segment (see S4 Fig), but can also be the result of cell death, the presence of debris, or movement of cells out of the field of view. The linkage error rate seems to be lowest for the 2D U-Net model trained on the high-cell-density data, which also had shown nominally better segmentation performance. Using the high-cell-density U-Net model, the linkage error rate as a function of track length is shown in Fig 4D. We observe that tracks longer than 9 hours converge to the error rate of 0.005 –indicating higher confidence in longer tracks. Track length distributions are shown in Fig 4E. Tracking the GFP reference data produces more long tracks than the 2D U-Net-derived inferenced data as expected; however, we see that when applying the high-cell-density model, the label-free segmentation results are the most similar to the GFP results, with the high-cell-density model having the longest tracks. Comparing the inferenced tracking results to the GFP tracking results (and assuming GFP tracking is 100% accurate), we can calculate the relative number of cells that were tracked for the entire 22 hours for the different cell-density models: for the high-cell-density, medium-cell-density, and low-cell-density models, the relative long tracks were 79%, 52% and 42% respectively. Finally, Fig 4F shows the interdivision times determined from the different cell-density models and from the GFP fluorescence data, which all had a median around 15 h. We make three observations based on these results: (1) the results from the high-cell-density model most closely matches the GFP fluorescence reference dataset, (2) we track more cells from division to division with the high-cell-density model, and (3) the low-cell-density segmentation model results in interdivision times less than 9.5 hours–the lower limit of biologically realistic interdivision times [22]. This result suggests that using the low-cell-density segmentation U-Net leads to more tracking errors than the high-cell-density model. Accordingly, we use the high-cell-density U-Net for the rest of this study.

It is interesting to note that all models performed similarly with respect to segmentation over 20 h in culture, from the time in culture when cells are relatively sparse and later in time when they are at higher densities as shown in Fig 2C. Regardless of the cell density that the models were trained on, all of the models performed similarly when inferencing samples at lower cell densities and at higher cell densities. One difference between the 3 models was the number of cells that were provided for training. While the total number of pixels used for training the 3 models was kept constant, the number of cell objects in the training sets were 38875, 80997, 214944 for the low-cell-density, medium-cell-density and high-cell-density models, respectively, suggesting that the U-Nets are sensitive to number of objects used in training.

In addition to the data shown in Fig 4 which were extracted from a dataset collected over 22 h, we also examined data that were collected on cells that were more sparsely seeded initially and tracked for **36 h**. These data are shown in S4 Video. Shown in S8 and S9 Figs are the cell count and the mitosis rate, respectively, as a function of time in culture. It is noteworthy that mitosis rates did not vary significantly over the 36 h, indicating that mitosis rate is apparently not affected by higher cell density at longer times in culture. A distribution of interdivision times is shown in S10 Fig, which indicates the lifetime of cells between its birth and its subsequent division. The median division time was approximately 15 h.

## Temporal and spatial analysis of mitosis and migration

Previous studies have demonstrated position-dependent differences between human iPSCs when examining cells at colony edges versus colony centers. Differences in phenotypic characteristics (e.g. focal adhesion attachments) [23–25] and differences in functional responses (e.g. differentiation potential) [26–28] have been observed. These previous studies are limited to

single timepoint measurements or averaged responses; the label-free segmentation, mitosis detection and tracking described in Figs 2–4 above can be used to quantify spatiotemporal characteristics of many single iPSCs in live cell populations.

The size, shape, and density of colonies vary across the culture substrate. The data in Fig 5 examine whether dynamic single cell characteristics, namely the extent to which cells divide and move, are associated with the location of cells relative to the colony edge. Each colony was segmented according to the distance from the edge of the colony (Fig 5A). As cells grow and divide, each colony increases in size and cells on the interior are increasingly further from the edge. The mean squared displacement (MSD) measured for each individual cell is used to characterize how much a cell moves. There is no clear dependence of cell motion on experiment time, but the motion of individual cells varies spatially within the colonies (Fig 5B). Individual cells in the center of the colony tend to move less than cells near the edge. The mean of the MSDs calculated for motion over one hour time frames is plotted as a function of distance from colony edge in Fig 5C and varies from 450 μm² to 140 μm². On average individual cells move the most at the colony edge. Movement decreases as a function of distance from the edge up to 150 μm, when the average movement becomes constant with increasing distance from the edge. This can be observed in the heatmap overlayed on the colonies in Fig 5B. While this might not be a surprising result, the large number of cells that are included in this analysis (approximately 10,000) provides confidence in this conclusion. Average mitotic rates, on the

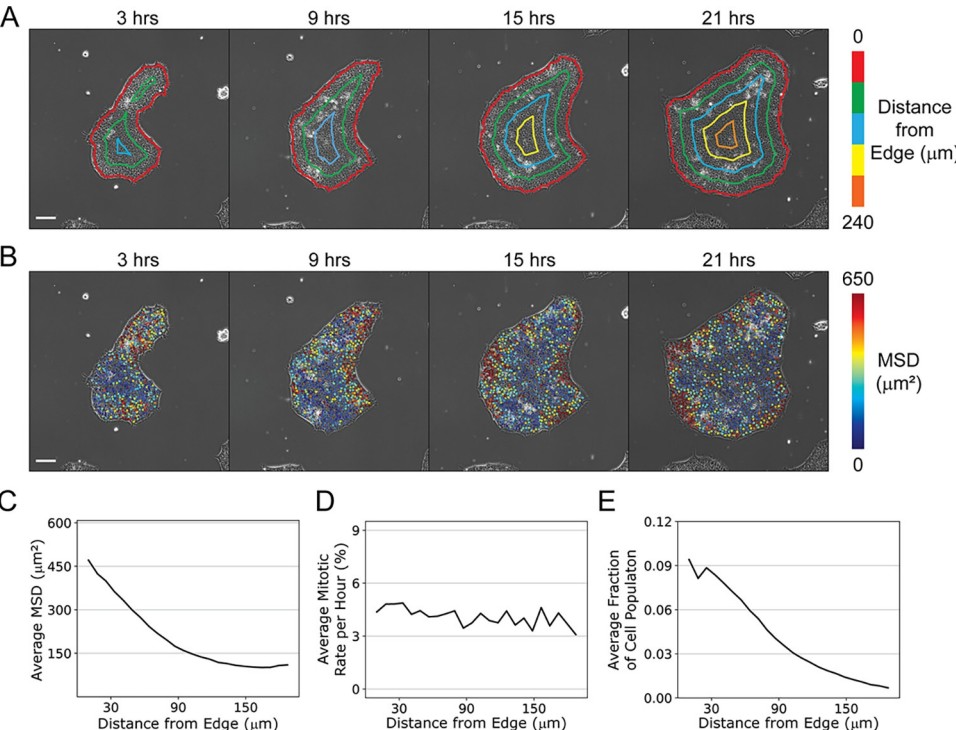

**Fig 5. Effect of spatial location within colonies.** (A) Phase images of a colony at the times in culture indicated with distances from the edge of the colony labeled with contour lines (scale bar = 100 μm). (B) Images shown in (A) but each cell is colored by its mean square displacement (MSD) calculated over the subsequent hour (scale bar = 100 μm). (C), (D) and (E) show the MSD, mitotic rate and cell fraction as a function of distance from edge of colony, respectively, computed over a time-lapse image series approximately 2400 x 2400 μm over 24 h. Cells move more at the colony edges, but their mitosis rate is unaffected. (Tracks < 100 minutes were not included in this analysis. The number of cells for this analysis increased from 4065 to 10811 over the course of the 24 h of imaging.) No systematic difference in MSD was observed over the time course of the experiment.

other hand, do not appear to depend on distance from the colony edge (Fig 5D), indicating that while cells are more migratory near the edges of the cells, they are not dividing at a higher frequency.

## Effect of excitation light exposure on cellular mitosis and cell number

Automated training data for the U-Nets was generated from images of cells with fluorescent nuclei. To confirm that the cells that were exposed to fluorescence excitation light were not significantly different from normal healthy cells, we examined their growth characteristics and compared them to cells that were subjected only to the transmitted light for phase contrast imaging. As shown in Fig 6A, exposure of cells to the minimal intensity of fluorescence excitation light (56 mJ/cm$^2$ referred to as 1x) had little effect on cell number over time compared to cells in wells that were only exposed to transmitted light (indicated by the gray line). This result suggests that cells were minimally perturbed by the amount of light they were exposed to at 2-minute intervals over 20 hours. The corresponding data in Fig 6B also indicate that the mitosis rate was apparently unaffected. In contrast, in wells that were exposed to increasing amounts of excitation light, the rate at which cell number increased over time is considerably perturbed. These data are shown in Fig 6A where relative excitation light is indicated in green text as 1.4 x, 2.1x, and 3.6x the minimum light exposure that was used for generating reference data. These higher intensities of excitation light led to significant cell death that was apparent by manual inspection of images, and by the reduced relative cell numbers as shown by the green lines (example colonies shown in S5 Video). This was not surprising since radiation is a known inducer of apoptosis [29].

Interestingly, Fig 6B shows that the number of mitotic cells relative to total cells in the wells was approximately equivalent for wells that were exposed to fluorescence excitation light at any intensity compared to wells that were not exposed to excitation light. The data in Fig 6A show that there are fewer cells in the wells exposed to the higher intensity of fluorescence excitation light relative to the untreated wells, suggesting that many cells died as a result of exposure to higher intensities. The smaller number of remaining cells apparently continued to divide at a normal rate as indicated by the data in Fig 6B. This result is similar to studies that have shown that chemical induction of apoptosis results in a wide variability in the extent and rate of cell death within isogenic populations of different cell types [29–31]. A comparison of the distribution of division times in cells that were either treated or not treated with an inducer of apoptosis resulted in no significant effect on cell cycle times [29].

The distinction between mitosis rates and cell counts in response to higher intensities of irradiation is evident in Fig 6C and 6D. Fig 6C shows apparent doubling times determined from cell counts. Fig 6C shows that doubling times for cells exposed to the lowest intensity of fluorescence excitation light (shown in the first dataset) are very similar to that of unexposed cells, and the doubling times for unexposed cells over the four different datasets are all similar. However, for the three datasets exposed to higher fluorescence excitation light intensities, the cell count and the doubling times calculated from cell counts, shown in green, indicate a strong adverse effect of these higher intensities of light exposure.

It is noteworthy that there is a broad range of doubling times calculated over 60-minute intervals, which is evidence of the variability in susceptibility to the effects of light exposure on apoptosis within the populations. This variability is borne out when calculating doubling times based on mitotic events. In the remaining cells in the culture, i.e., the cells that were not undergoing apoptosis (Fig 6D), the doubling times are similar to those for unexposed cells. This observation of a very small effect of light exposure on measured rates of mitosis is consistent with previous studies on HeLa cells [29]. Manual examination of images confirmed that

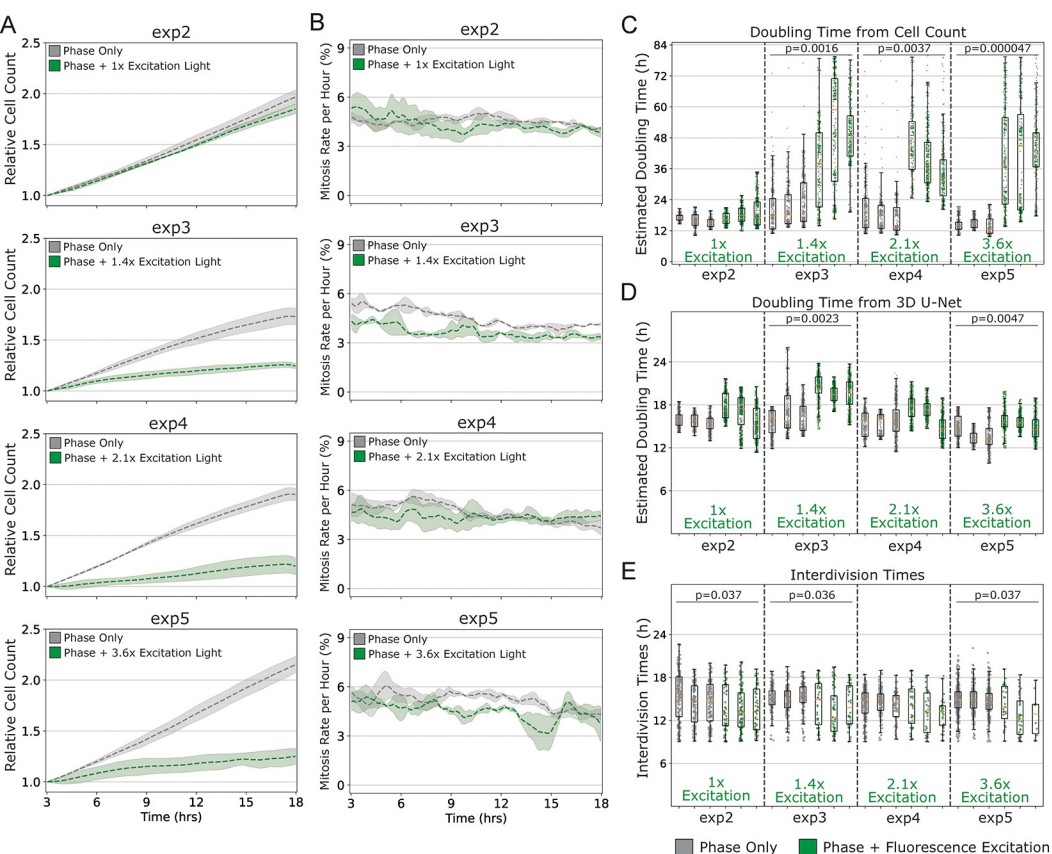

**Fig 6. Effect of excitation light exposure on cell division.** (A) Cells were exposed only to transmitted light (indicted by grey lines) or to transmitted light and excitation light (indicated by green lines) for 1 sec at 2 min intervals. The relative excitation light is indicated in each panel; 1x is the level of excitation light at which training data were acquired; 1.4x, 2.1x, 3.6x refer to the fold increase in excitation light to which cells were exposed. The number of cells is reported over time relative to the number of starting cells. Shadows around lines reflect standard deviations from replicate samples (n = 3). (B) Mitosis rates were determined from the samples analyzed in 6A using the 3D U-Net mitosis detector to determine the total number of mitotic events that occurred within 60 min divided by the initial number of cells in the image in a moving 2-minute time window followed by a 20-frame moving average to reduce noise. (C) Tukey box plots of doubling times calculated from the cell count data corresponding to the relative exposure levels shown in each panel of 6A. Each data point represents a doubling time calculated every two minutes from the numbers of cells in 30 sequential frames (60 min). For the samples exposed to relative light levels that were greater than 1x exposure (green indicators), doubling times are significantly longer than for the other samples. Doubling Time = $\ln(2)/(\ln(N_t/N_0)/t)$. (D) Tukey box plots of doubling times calculated from the numbers of mitotic events determined in samples shown in each panel of Fig 6B corresponding to their relative exposure levels. Doubling Time = $\ln(2)/(\ln(N_{t0}+\#mitoses)/N_0)/t)$. (E) Tukey box plots of interdivision times determined from the samples that were subjected to the relative light exposure indicated. Each dot represents the interdivision time for a single cell. The data associated with exposure to higher levels of excitation light resulted in very few cells that progressed from one cell cycle to another, although interdivision times were not different. All data are based on a stitched image of 2x2 fields of view with a minimum track length of 100 minutes. All statistical tests were unpaired t-tests using the mean of each replicate comparing within the same experiment.

mitotic cells were accurately identified by the 3D U-Net in these samples. The conclusion from these data is that, over the time course of our observation, high amounts of light will kill some cells in the population while apparently having little to no effect on the rates of division of other cells. Interdivision times in these populations (Fig 6E) show similar results for all samples, but for samples that were exposed to higher light intensity, many fewer cells are tracked for the length of an interdivision time.

These results are an example of the power of quantitative live cell image data. In the absence of cell-scale data, the low proliferation rate of the population of cells exposed to increasing

amounts of light would indicate an apparent phototoxic effect that may be associated with increased death and /or decreased cell division. But the additional mitosis, individual cell distribution, and temporal metrics indicate more nuance in the relationship between mitosis, migration and phototoxicity.

## Discussion

The results of this work allow the use of phase contrast images of iPSCs to provide quantitative cell characteristics, such as mitotic rates, cell migration rates and other dynamic behaviors on a cell-by-cell level and with spatial and temporal resolution. Phase contrast, compared to bright field microscopy, provides high contrast images from which spatial information can be extracted. For example, we developed a 3D U-Net to detect mitosis using information from both phase and fluorescence images. Data are acquired from single image planes, and therefor can be a more efficient modality. By using phase contrast images for locating and tracking individual cells, fluorescent reporters within cells can be probed intermittently, allowing control over cell exposure to excitation light. Furthermore, the use of phase contrast imaging for tracking cells over their lifetime and the lifetimes of their progeny means that cell lineages can be tracked even if cells are not always expressing the reporter. It will be possible to query a very large number of individual cells to study the heterogeneity in temporal responses of cellular features within a population, and the correlations in fluctuations of different features in individual cells. This approach may, for example, provide insight into gene regulatory networks [32]. In addition, access to spatial information of individual cells permits the assessment of the effects of changes in culture conditions on cell-cell interactions, the relationship of location within the colony to cell division and motion. For example, these data, made possible by the spatial and temporal capabilities of the quantitative imaging and image analysis pipeline, suggest that migration rate and mitosis are not correlated, and that cells do not divide any more frequently if they reside at the edge of a colony than if they are in the middle of a colony. The large number of cells in this analysis provides confidence in this result. We also demonstrated the surprising observation of photo-induced toxicity of some cells in a population, while the mitosis rates in other cells appeared to be normal.

In this work, we have acquired and analyzed very large image datasets of hundreds of thousands of cells, which will be required for uncovering multiparameter correlations in cell populations. Large datasets are also required for the development of reliable models for image analysis, and we demonstrate here the use of fluorescent nuclear probes to automate the process of creating training data. The use of automated segmentation to identify the nuclear foreground from the non-nuclear background allows the creation of very large training and reference data, removing a serious limitation associated with manual annotation. By increasing the number of cells that go into training, we increase the probability that full representation of the heterogeneity of the cell population will be achieved. Similarly, reference data for assessing U-Net model accuracy were also created by automated segmentation of fluorescent images. In addition, having large datasets to analyze provides confidence in the reliability of the results.

We have demonstrated that we can, with confidence, use reference data that were generated with classical algorithms to compare models. The 2D U-Net model for nuclear segmentation was trained from single time point images, a process that can be easily and quickly replicated if needed. Rapid data acquisition also allows us to image many fields of view and many cells in phase contrast at 2 min intervals; this rapid rate helps the accuracy of cell tracking, allowing for a tracking efficiency of approximately 80% of nuclei over 22 h of tracking time. In comparison, using an AI algorithm, Atwell et al. reported 50% of nuclei accurately tracked for about 3 h when acquiring bright field images at 7 min intervals [14].

The 3D U-Net model which identifies classes of mitosing cells requires time-lapse images of sufficient frequency; more time-lapse training data would be required to achieve the same segmentation accuracy from the 3D U-Net model as the 2D U-Net model.

We have demonstrated some examples of the sensitivity of parameters used in preparing the data and selecting the models to the inferenced outcome, and some of the caveats are discussed in S1 Text. We have clearly not examined all the possible parameters that could be explored. Complicating the analysis process is the variability that is typical in biological samples from one preparation to another. Despite attempts to be consistent, clearly there are subtle unknown differences in cell and material handling that result in different culture behaviors, particularly from one day to another. These variations may be convoluted with operation of the models, and occasionally there are culture conditions that preclude satisfactory application of any model, such as floating debris that obscures underlying cells. In addition, one might want to create filters to recognize bad frames where, for example, cells were occluded by floating debris. For example, we found that applying temporal filters to remove objects that are tracked only for short times is a strategy for examining only those cell objects that are tracked with a very high level of confidence. The use of a temporal filter to eliminate shorter tracks can be applied selectively depending on the experimental objectives. For example, if a large statistical sampling was desired for determining the rate of population expansion, one might use a less stringent filter than if the long-time fluctuation in cellular expression of a gene reporter were of interest.

Ultimately, the use of large volumes of reliable quantitative real time images will aid not only cell biology research, but also the monitoring and evaluation of cell manufacturing processes. A recent report demonstrates the use of imaging and AI analysis for in-line monitoring to reduce variability and improve quality control of iPSC cultures destined for cell therapies [33]. Especially for manufacturing applications, image analysis must be rapid, automated, and demonstrated to be reliable. We have developed an image acquisition and analysis pipeline that can be efficiently trained and inferenced on a new dataset. Hundreds of thousands of cells for training are imaged in 10 min, and training and inferencing takes a few hours. Generalizability is also important for adapting the model to different applications. While we used a GFP-producing reporter cell line for training and reference data, one could use different nuclear (or other) markers for model development. We have not examined generalizability exhaustively in this study, but we did observe that the 2D U-Net model performed well on other cell lines, namely the parental WTC-11 cell line and a human embryonic stem cell, H9, using a different fluorescent nuclear probe for reference.

## Conclusion

In this work we have established that single cells in iPSC colonies can be identified and tracked for long times with good accuracy, and that automated segmentation and classification of nuclei based on fluorescent probes can obviate the need for manual annotation. The use of automated annotation in combination with high-speed acquisition rates will enable the efficient development of multiple models to analyze cells in different biological states and conditions over time. This is important for creating useful AI models with high accuracy, and it allows the sampling of many cells as required to capture cell-to-cell variability. This capability will provide insight into complex biological processes by enabling the quantitation of correlations between many features of iPSC systems that are accessible by real-time imaging, and lead to better understanding of the controlling factors of emergent cellular properties and how to predict and direct them. In addition to the cellular characteristics, we have examined here (motility, division rates and interdivision times), this work will facilitate the monitoring of many metrics of interest over space and time, including intracellular structural features,

expression of transcription factors and other markers of gene expression, and markers of functional response such as differentiation.

## Supporting information

**S1 Text. Supplementary methods and caveat discussion.**
(PDF)

**S1 Table. Summary of experiments and experimental conditions used.**
(XLSX)

**S1 Video. Timelapse images showing representative output of the 3D U-Net.** The field of view shown is the same as shown in S3 Video for comparison. The class outputs are as follows: nucleus = purple, mitotic nucleus = orange, daughter nucleus = yellow, and background = black. Time is shown as h:min.
(MP4)

**S2 Video. Time-lapse images of exp0 showing the phase contrast channel (greyscale) and overlayed with the inferenced and tracked nuclei (color).** Individual tracked cells are identified as objects that maintain the same color from one frame to the next. The initial and final cell counts were 3983 and 11358, respectively. Debris on the detector can be seen to create occasional spurious nuclear objects by the 2D UNET; detecting such imperfections in the imaging system and correcting them is important for minimizing inaccuracies. Upon removal, the generation of spurious objects due to the debris was eliminated. Time is shown as h:min.
(MP4)

**S3 Video. Sub-region of S1 Video shown with higher spatial resolution.** The phase contrast channel (greyscale) is overlayed with the inferenced and tracked nuclei (color). Individual tracked cells are identified as objects that maintain the same color from one frame to the next. The movie begins with two colonies in the center of the field of view. Time is shown as h:min.
(MP4)

**S4 Video. Cells imaged continuously for 36 h.** Cells were seeded at a relatively low density to permit longer term imaging.
(MP4)

**S5 Video. Selected timelapse phase contrast images of iPS cells exposed to varying levels of fluorescence excitation.** From right to left: 0x, 1x, 1.4x, 2.1x and 3.6x light dose. Time is shown as h:min.
(MP4)

**S1 Fig. The effect of relevant Fogbank algorithm parameters on the F1 score are evaluated.** The Fogbank algorithm is applied after the 2D U-Net to separate two or more nuclei that share a boundary and are considered one object. The 'erode_size' parameter is varied from 1 to 5 and for each 'erode_size' value, the 'min_size' parameter is varied from 5 to 15. The highest F1 scores for segmentation accuracy can be obtained with 'erode-size' in the range of 1 to 4 and 'min_size' in the range of 8 to 10.
(PDF)

**S2 Fig. Representative sub-image from exp0 showing variability associated with training and inferring with the 2D U-Net.** The color scale indicates the number of times a trained U-Net inferred that a pixel was classified as a nucleus. Many pixels exhibit high model concordance (9/9), while other pixels exhibit larger discordance. Scale bar = 25 μm.
(PDF)

**S3 Fig. Inference performance when a threshold of multiple models is applied.** A threshold was applied to the image data in exp0 (a representative sub-image shown in S2 Fig) and the model performance scores: 'Fraction of Missing Objects', 'Fraction of Additional Objects' and 'F1 Score' are plotted as a function of the threshold value. As expected, the 'Fraction of Missing Objects' increases with threshold value, the 'Fraction of Additional Objects' decreases with threshold value, and the 'F1 Score' is highest for intermediate threshold values.
(PDF)

**S4 Fig. Concordance between nuclei detected manually, by automated analysis of GFP fluorescence image, and by AI of the phase image.** Three sets of detected nuclei data are shown: Nuclei detected by manual inspection of GFP fluorescence images (red dots), nuclei detected by classical image analysis of GFP fluorescence images (green dots) and nuclei detected by AI-based analysis of the phase contrast images (yellow dots). Scale bar = 25 μm. Many image regions illustrated high concordance between the three datasets, whereas the inset square highlighted in orange illustrates a region of high discordance (scale bar = 10 μm). The GFP fluorescence-based automated image analysis tends to merge nuclear objects compared to the manual annotations and the AI-based analysis of the phase contrast images.
(PDF)

**S5 Fig. Model performance evaluated based on manual versus automated methods.** The model performance scores: 'Fraction of Missing Objects', 'Fraction of Additional Objects' and 'F1 Score' are plotted for inferenced data from 4 different U-Net models: the low-cell-density model, the medium-cell-density model, the high-cell-density model and a model trained on a mixture of data from all cell densities. The reference data for computing the scores was derived from either nuclei detected by inspection of the GFP fluorescence images or the nuclei detected by classical image analysis of the GFP fluorescence images. The relative performance of the models is similar using either approach for generating reference nuclei.
(PDF)

**S6 Fig. Model performance on other cell lines.** Using the high-cell-density model, the F1 scores are plotted for both H9 hESC (exp6.0) (F1 = 0.89) and the parental WTC11 lines (exp6.1) (F1 = 0.90). Each datapoint represents the corresponding error rate for that frame, and the dot color indicates the density of cells in the frame. Tukey box plots indicate summary statistics for each timelapse dataset.
(PDF)

**S7 Fig. Example of track filtering based on minimal track length.** Green dots represent tracks that have a track length greater than the minimum track length time and are kept. Red dots represent tracks that are filtered out due to having a track length smaller than the minimum track length time. Corresponding phase and fluorescence images are shown for minimum track length times of 0, 1, 2, and 4 hours. Scale bars = 25 μm.
(PDF)

**S8 Fig. Cell count over 36 h in culture.** Number of nuclei imaged as shown in S4 Video at each time point during 36 h.
(PDF)

**S9 Fig. Rate of mitosis during 36 h in culture.** Number of mitotic events that occurred in a 30 min time frame divided by the cell count during that time frame plotted over 36 h.
(PDF)

**S10 Fig. Distribution of interdivision times.** The number of interdivision times (cell life-times) determined by tracking of inferenced phase images plotted in 1-hour bins to show the distribution of cell lifetimes (n = 420).
(PDF)

## Acknowledgments

**Disclaimer:** Commercial products are identified in this document in order to specify the experimental procedure adequately. Such identification is not intended to imply recommendation or endorsement by the National Institute of Standards and Technology, nor is it intended to imply that the products identified are necessarily the best available for the purpose.

## Author Contributions

**Conceptualization:** Michael Halter, Anne L. Plant.

**Data curation:** Anthony J. Asmar.

**Formal analysis:** Anthony J. Asmar, Zackery A. Benson.

**Investigation:** Anthony J. Asmar, Zackery A. Benson.

**Methodology:** Anthony J. Asmar, Zackery A. Benson, Adele P. Peskin, Michael Halter.

**Project administration:** Michael Halter, Anne L. Plant.

**Software:** Anthony J. Asmar, Zackery A. Benson, Adele P. Peskin.

**Supervision:** Michael Halter, Anne L. Plant.

**Visualization:** Anthony J. Asmar, Zackery A. Benson, Mylene Simon.

**Writing – original draft:** Anthony J. Asmar, Zackery A. Benson, Michael Halter, Anne L. Plant.

**Writing – review & editing:** Anthony J. Asmar, Zackery A. Benson, Adele P. Peskin, Joe Chalfoun, Michael Halter, Anne L. Plant.

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
