## [Decision Letter · Decision Letter 0]

26 Nov 2023

PONE-D-23-34339High-volume, label-free imaging for quantifying single-cell dynamics in induced pluripotent stem cell coloniesPLOS ONE

Dear Dr. Asmar,

Thank you for submitting your manuscript to PLOS ONE. After careful consideration, we feel that it has merit but does not fully meet PLOS ONE’s publication criteria as it currently stands. Therefore, we invite you to submit a revised version of the manuscript that addresses the points raised during the review process.

We look forward to receiving your revised manuscript.

Kind regards,

Zhiming Li, Ph.D.

Academic Editor

PLOS ONE

Journal Requirements:

Reviewers' comments:

Reviewer's Responses to Questions

**Comments to the Author**

1. Is the manuscript technically sound, and do the data support the conclusions?

Reviewer #1: Yes

Reviewer #2: Yes

2. Has the statistical analysis been performed appropriately and rigorously? 

Reviewer #1: N/A

Reviewer #2: Yes

3. Have the authors made all data underlying the findings in their manuscript fully available?

Reviewer #1: Yes

Reviewer #2: Yes

4. Is the manuscript presented in an intelligible fashion and written in standard English?

Reviewer #1: Yes

Reviewer #2: Yes

5. Review Comments to the Author

Reviewer #1: The authors proposed a new segmentation approach with 2D/3D U-net for human pluripotent stem cells in this study. Using a GFP-labeled nuclear tag, they trained U-Net to detect mitosis. Overall, it was good to demonstrate their algorithm tracking iPSC nuclei for ~20hrs with label-free imaging. However, there are concerns.

The major concerns are (1) novelty, (2) short tracking period, (3) applicability

(1) Recently major microscopies come with AI-based imaging support. Thus, I’m not intrigued by the authors’ approach in terms of novelty or performance. The algorithm is specialized to iPSC culture, does this really outperform others?

AI-based imaging supports are often proprietary, so we don’t exactly know how they work. There are also open algorithms available. For instance, Bright2nuc was proposed (Atwell, Cell Rep Methods, 2023) to work even in 3D structures. It is important to discuss the advantages of the proposed approach over other approaches, and how it sits in the current perspective.

(2) The authors tracked cells only 20 to 21hrs, which sounds short to me. I would like to see at least 48 hrs, which should cover two cell divisions. It is interesting that cell proliferation evenly occurs regardless of the location within the colony (Fig 5). If the authors can track cells longer, then the authors address whether the proliferation weans or remains the same throughout the culture period. If it can apply for much longer, the approach can be applied for differentiation studies, too.

(3) Establishing an algorithm is important, and it is more important that the algorithm can be applied to any given system/image. Is it possible? It should be at least discussed, and if possible, please demonstrate if the algorithm works on images from different microscopes/different imaging settings.

(4) In Figure 6, the authors demonstrate that fluorescence excitation reduced cell number due not to reduced proliferation but to increased apoptosis. It wasn’t unclear if the authors were able to detect such cells in this approach. Can it be dictated by the likes of the drop-out rate? Is it only detectable with manual inspection (line 537)?

(5) Something that I couldn’t get was a “fraction of additional objects”. What actually are the “additional objects”? In addition, in Figure, it was labeled as “extra objects”.

(6) The figures mainly displayed the training and confirmation process. I would like to see more demonstrations of what the algorithm can do/apply.

Reviewer #2: High-Volume, Label-Free Imaging for Quantitative Single-cell Dynamics in Induced Pluripotent Stem Cell Colonies

• Overall Thoughts

o In this manuscript, the authors demonstrate a machine-learning method for nuclear segmentation, cell tracking and mitosis detection in phase contrast (PC) imaging of iSPC colonies. Due to the toxicity of prolonged fluorescent imaging, such a workflow would allow for long-term quantification of cell state and dynamics over much longer periods than otherwise feasible. Additionally, the issue of photobleaching often impacts experimental design, irrespective of cell health, limiting the scope in some instances. Circumventing these issues, at least in this specific context, would present a significant step forward in microscopy experimental design. Indeed, the capacity for AI to help decrease the amount of illumination necessary in experiments remains one of the largely untouched areas in microscopy. This work offers a step in this direction. Moreover, by developing a model which considers the information contained in PC images, this work points towards a class of models built upon this dye-free ground truth.

o Their proposed algorithm takes the form of linked 2D and 3D U-Nets trained on fluorescent data for nuclear segmentation. The authors evaluate the accuracy of their model by testing it at multiple cell densities and cell lines, suggesting the model is plausibly generalizable. Moving forward, it will be interesting to see whether such a workflow functions on other cell types and in other imaging contexts. Further, they use this model to offer preliminary quantification of cell motility and more importantly, the mal effects of fluorescent imaging on colony health, suggesting this form of analysis can elucidate the details of this toxicity. The authors also assert they envision this pipeline facilitating the study of gene expression in a label-free environment.

o In total, this well-written manuscript presents an imminently useful pipeline for tracking and quantifying iPSC colony behavior dye-free.

• Review Criteria

o Originality

This work builds on the authors’ previous models by combining and optimizing pieces of other pipelines into a cohesive workflow for segmentation, tracking and cell state determination. As far as can be discerned, this work does not appear in any other publication.

o Technical Standards

The statistical significance of quantified results is effectively measured and communicated. In the more qualitative sections, the authors clearly describe differences and show, to the extent possible, their reasoning through their data.

• The one instance I would like to see extended and more robust reasoning is the assertion that the pipeline functions best at the highest colony density, as explained below.

o Interpretation

The data present clear behaviors from which the authors derive reasonable and justified conclusions.

o Presentation

The manuscript is well-written and structured.

o Ethics

No issues are readily apparent.

o Data availability

The data and code are all available through the stated repository, and the code itself is sufficiently documented.

• Specific Comments

o Throughout, the authors indicate (and the data demonstrate) that the pipeline’s accuracy is best when applied to the highest density colonies. I would appreciate and suggest a discussion of the authors’ interpretation and intuition regarding this fact. It is entirely possible there is a biological reason that I am missing, but if there is an algorithmically relevant reason, that could be discussed. Additionally, there does not seem to be the same trend in accuracy as a function of density when plotted in Fig 2C. If anything, the F1 score decreases as the density increases. I would suggest the authors elaborate on these ideas.

o In Fig4, the authors show the effect of track length filtering on model accuracy. As the minimal track length increases, the number of spurious detections decreases, but the number of missed detections increases. This suggests some ideal track length—indeed the plot of F1 score as a function of minimal track length seems to display a rough peak of sorts. I would be interested in the authors’ thoughts on whether such an ideal threshold exists, and if so, what that might mean physically.

Additionally, while the data do suggest that pipeline functions best at highest colony density, this conclusion would be strengthened by a clear singular measurement of this fact. For example, if the authors could quantify by how much better the pipeline performs, that could be useful.

o Fig5 represents what I consider to be the manuscript’s weak link. I appreciate and respect that the authors want to demonstrate the pipeline’s usefulness in answering physical questions. In that vein, these data are sufficient. However, this section reads as rather thin when compared to the totality of the work. In other words, the main conclusion from this figure—that cells move more at the colony periphery than the center—is so unsurprising it falls flat as a demonstration of the algorithm’s use.

Instead, I would move Fig6 before this one. Highlighting the issue of phototoxicity and how this pipeline serves to mitigate it strikes me as much more impactful than the result(s) of Fig5. In fact, I would suggest the authors bring in this idea to Fig5, possibly in showing the effect of illumination on cell motility. This would both demonstrate that the pipeline facilitates such measurements, but also connects it to what has come before.

o There are a few minor elements of Fig6 I would suggest that the authors change. The titles of the plots in A and B do not effectively communicate that they represent different illumination amounts. I would make this clear to the reader. Similarly, this applies to the individual boxes in the turkey box plots as well.

o Otherwise, outside a relatively small number of minor proofreading errors, the manuscript is well-written and structured.

6. PLOS authors have the option to publish the peer review history of their article (what does this mean?). If published, this will include your full peer review and any attached files.

Reviewer #1: No

Reviewer #2: **Yes: **Owen Puls

---

## [Author Response · Author response to Decision Letter 0]

10 Jan 2024

Response to reviews has been attached as a separate file.

---

## [Decision Letter · Decision Letter 1]

24 Jan 2024

High-volume, label-free imaging for quantifying single-cell dynamics in induced pluripotent stem cell colonies

PONE-D-23-34339R1

Dear Dr. Asmar,

We’re pleased to inform you that your manuscript has been judged scientifically suitable for publication and will be formally accepted for publication once it meets all outstanding technical requirements.

Kind regards,

Zhiming Li, Ph.D.

Academic Editor

PLOS ONE

Additional Editor Comments (optional):

Reviewers' comments:

Reviewer's Responses to Questions

**Comments to the Author**

1. If the authors have adequately addressed your comments raised in a previous round of review and you feel that this manuscript is now acceptable for publication, you may indicate that here to bypass the “Comments to the Author” section, enter your conflict of interest statement in the “Confidential to Editor” section, and submit your "Accept" recommendation.

Reviewer #1: All comments have been addressed

Reviewer #2: All comments have been addressed

2. Is the manuscript technically sound, and do the data support the conclusions?

Reviewer #1: (No Response)

Reviewer #2: Yes

3. Has the statistical analysis been performed appropriately and rigorously? 

Reviewer #1: (No Response)

Reviewer #2: Yes

4. Have the authors made all data underlying the findings in their manuscript fully available?

Reviewer #1: (No Response)

Reviewer #2: Yes

5. Is the manuscript presented in an intelligible fashion and written in standard English?

Reviewer #1: (No Response)

Reviewer #2: Yes

6. Review Comments to the Author

Reviewer #1: (No Response)

Reviewer #2: The authors adequately addressed my specific concerns as raised in my initial review. I still disagree with the authors on the relative impact of the data presented in Figure 5 and maintain the submission would be stronger if the order of Figures 5 and 6 were reversed and/or combined/modified in some fashion. However, I take the authors' rebuttal seriously and understand their reasoning. Outside of this, I consider this work to represent a meaningful and significant contribution to the problem of segmentation and tracking.

7. PLOS authors have the option to publish the peer review history of their article (what does this mean?). If published, this will include your full peer review and any attached files.

Reviewer #1: No

Reviewer #2: **Yes: **Owen Puls

---

## [Editor Report · Acceptance letter]

12 Feb 2024

PONE-D-23-34339R1 

PLOS ONE

Dear Dr. Asmar, 

I'm pleased to inform you that your manuscript has been deemed suitable for publication in PLOS ONE. Congratulations! Your manuscript is now being handed over to our production team.

Kind regards, 

on behalf of

Dr. Zhiming Li 

Academic Editor

PLOS ONE